# Aesthetic judgment of architecture for Chinese observers

**Anbang Dai** [1], **Jiajie Zou**[2], **Junru Wang**[1], **Nai Ding**[2], **Hiroatsu Fukuda**[1]*

**1** Department of Architecture, The University of Kitakyushu, Kitakyushu, Japan, **2** College of Biomedical Engineering and Instrument Sciences, Zhejiang University, Hangzhou, China

* fukuda@kitakyu-u.ac.jp

**Data Availability Statement:** All relevant data are within the manuscript and its Supporting information files.

**Funding:** The author(s) received no specific funding for this work.

## Abstract

Architects should consider the aesthetic experience of potential users when designing architectures. Previous studies have shown that subjective aesthetic judgment of architectures is influenced by structure features, and Western observers prefer structures that have curvilinear contours, high ceilings, and open space. The building styles, however, vary across cultures, and it remains unclear whether the preference for contours, ceiling height, and openness exist across cultures. To investigate this issue, this study analyzes the aesthetic judgment of Chinese observers, and the results demonstrate that Chinese observers also prefer high ceilings and open space. Preference for curvilinear contours, however, interacts with ceiling height and openness. Simple effect analysis reveals that Chinese observers prefer curvilinear contours only when the ceiling is low and the space is closed. In sum, these results suggest that preference for high ceilings and open space is robust for Chinese observers, but the preference for curvilinear contours is less reliable.

## Introduction

Investigating the preference of architectural features from the perspective of empirical aesthetics allows architects to gather more information about how to design structures that can meet both functional and public aesthetic requirements. Environmental characteristics can trigger neurological and physiological responses in humans, thereby exerting a positive or negative impact on them [1–3]. To a certain extent, a good architectural design enhances users' comfort, cognition and creativity [4]. Architectural aesthetics connects emotion and aesthetics and strikes a balance between the two [5]. Previous studies have demonstrated the reward circuitry in the brain is activated when seeing artwork. Artists who know how to exploit this circuitry can intensify an individual's aesthetic experience [6]. Once a certain architectural element fits in a certain life scene, such as work, study, and rest, it can enhance behavioral effects through positive emotions [7, 8]. Currently, many architects have such ideas but lack the theoretical foundation as well as an understanding of the effect of some architectural factors on subjective experience.

In order to increase the understanding the relationship between architectural factors and subjective experience, researchers have done a lot of exploration in western culture. Studies in

**Competing interests:** The authors have declared that no competing interests exist.

the Western culture have shown that the aesthetic judgment of architecture is influenced by the response to specific sensory features, such as contour, ceiling height, and openness [1, 9]. Studies have showed that Western observers prefer structures with curvilinear contours, high ceilings, and open space [10–13]. Ceiling height and openness also impact people's perception and emotion [8, 14]. In structures with high ceilings, people tend to have more positive emotional responses, such as "happiness", "comfort" and "fun". Similarly, openness influences judgments of beauty and pleasantness, people tend to experience more positive emotions in spacious environments than in small environments [15, 16].

Although it has been demonstrated that contour, ceiling height, and openness are critical factors that influence the aesthetic judgment of architecture for Western observers, it remains unclear whether the preference to these features is universal. If the preference to architectural features is strongly influenced by daily architectural aesthetic experience, observers living in environments with different building styles may prefer different architectural features [17, 18, 22] Here, we analyze the preference to architectural features, including ceiling height, openness, and contour, in Chinese observers.

## Method

### Participants

The participants in this study were college students who were all right-handed, had no visual impairments and color blindness, had normal or corrected vision, and had no history of psychosis or neuropathy. The experimental protocol for this study was approved by the Research Ethics Committee of Zhejiang University School of Medicine (2019–047). Before the experiment, all participants signed a written informed consent form, and after completing the experiment, each participant received 40 RMB monetary reward. A total of 29 participants were included in this study, including 19 males (age: 23.05±1.99 years) and 10 females (age: 23.00 ±2.00 years).

### Stimuli and procedure

The current study followed the same procedure used in the study by Vartanian.et al. 2013 [10], and used the same stimuli. Twenty-nine participants were recruited in this study. Two hundred pictures of architectural space with different styles composed the stimulus material used this study. Each picture contained 3 factors, i.e., ceiling height, openness and contours, and each factor had 2 levels. Two hundred pictures of architectural space were classified high/low ceiling, open/closed space and curvilinear/rectilinear contours. Eight sets of pictures with different styles were generated to combine factors and factor levels, and each set contained 25 pictures, as shown in Fig 1. The pictures are from reference [10].

The experiment was conducted in a sound-proof booth. The participant used five buttons (1 to 5) on the left-hand side for scoring and one button (0) on the right-hand side for starting the test; the buttons were attached with rubber tape for easy finger recognition. On the edge of the table, there was an adjustable chin rest to enable the participant to rest his/her chin snugly on the chin rest after sitting and to fix his/her line of sight to the central upper quarter of the display screen.

Before the experiment, the instructor guided the participant to sit correctly, put his/her hands on the corresponding buttons and relax; the participant could adjust the height of the seat so that his/her chin could rest snugly on the chin rest. During the experiment, the participant was asked to keep his/her posture steady, without moving his/her chin. The instructor then left the booth, and the participant pressed the starting button (0) to start the experiment.

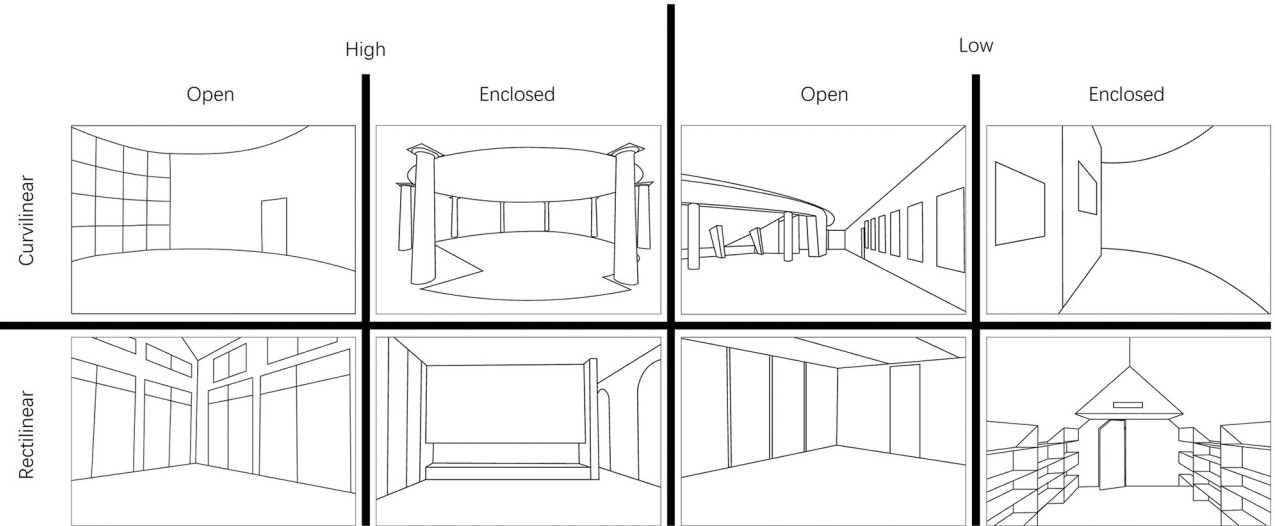

**Fig 1. Examples of stimuli.** A total of 200 pictures were divided into 8 conditions of 25 pictures each. The experiments used real photos and the line drawings are shown for illustrative purposes.

The experiment included 200 trials, and 1 picture was presented in each trial. The steps for each trial were as follows. First, a fixation point was shown in the center of the screen for 1 s. Second, a picture was displayed for 3 s, which was then followed by 2 questions that popped up on the screen, asking the participant to score the picture that was just displayed in terms of pleasantness and beauty (1 = very unpleasant/ugly; 5 = very pleasant/beautiful). The order of pictures was randomized for each participant. Play the next picture after the participant has scored the picture. The time to make pleasantness and beauty judgment was self-controlled and the total duration of the experiment was between 35 and 75 minutes. The research procedure is shown in Fig 2. The experimental program was written using the MATLAB 2018 psychtoolbox software [19].

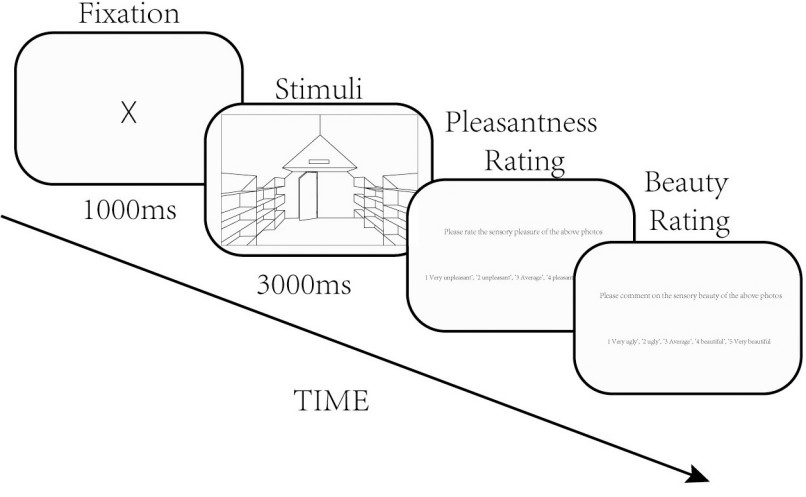

**Fig 2. Experimental procedure.**

## Data analysis

The experiment adopted a three-way repeated measures design, in which the 3 factors were ceiling height, openness and contour, which were all intra-group factors. Specifically, to examine the influence of ceiling height, openness and contour on the viewer's perceived pleasantness and beauty, as well as the possible interaction effect, three-way intra-group repeated measures analysis of variance (ANOVA) was adopted. First, for each participant, the sum of the scores for the 25 pictures in each set was calculated and treated as one "repeated measures" result for that participant (with a value range of 25–125 points). Because each participant was tested using 8 sets of pictures, 8 "repeated measures" results were obtained for each participant. Finally, the scoring results for all participants were used as response variables, and the 3 factors, i.e., ceiling height, openness, and contour, were used as intra-group factors in the repeated measures ANOVA model.

The three-way repeated measures ANOVA model included 3 main effect terms (ceiling height, openness, and contour), three two-way interaction terms (ceiling height × openness, openness × contour, and ceiling height × contour) and one three-way interaction term (ceiling height × openness × contour). First, the total variation was decomposed to set up an ANOVA table based on model structure, and then, the significance of the main effect and if the interaction effects of each factor was tested. If a three-way interaction item was statistically significant, then a simple-simple effect test was performed, i.e., under different experimental treatments of the combination of 2 factors, the influence of the remaining factor on the dependent variable was tested. The effect of multiple comparisons [20] were corrected using Bonferroni correction. All the data in this study were analyzed using the bruceR [21] package of R (version 3.6.3), and two-sided tests were performed, for which the significance level was set to $\alpha = 0.05$.

## Results

The participants separately rated the pleasantness and beauty of each architectural space after viewing it for 3 s. We first analyzed the beauty rating using 3-way repeated measures ANOVA (ceiling height × openness × contour). The 3 main factors significantly influenced the beauty (Table 1) and pleasantness ratings (Table 2). The beauty and pleasantness ratings are shown in Fig 3 and Table 3. Architectural space with higher ceilings were rated as more beautiful and more pleasant than architectural space with lower ceilings. Architectural space that featured open space were rated as more beautiful and more pleasant than architectural space with less open space. Furthermore, architectural space with curvilinear contours were rated as more beautiful and more pleasant than architectural space with rectilinear contours.

There was also a significant 3-way interaction between ceiling height, openness and contour. Simple-simple effect tests revealed that the observers always preferred higher ceilings and open space (Fig 4). However, ceiling height and openness modulated how contour influences

**Table 1. ANOVA results for beauty ratings.**

| Factors | MS | MSE | df1 | df2 | F-statistic | Uncorrected p-value | Corrected p-value | $\eta^2 p$ | $\eta^2 p$ 90% CI |
|---|---|---|---|---|---|---|---|---|---|
| Ceiling height | 3.65 | 0.05 | 1 | 28 | 75.05 | <0.001 | 0.007 | 0.728 | 0.571~0.815 |
| Degree of openness | 4.47 | 0.05 | 1 | 28 | 94.3 | <0.001 | 0.008 | 0.771 | 0.634~0.845 |
| Contour type | 0.25 | 0.02 | 1 | 28 | 10.56 | 0.003 | 0.017 | 0.274 | 0.068~0.473 |
| Ceiling height × Degree of openness | 0.02 | 0.01 | 1 | 28 | 1.21 | 0.281 | 0.05 | 0.041 | 0.000~0.213 |
| Ceiling height × Contour type | 0.05 | 0.01 | 1 | 28 | 3.24 | 0.083 | 0.025 | 0.104 | 0.000~0.302 |
| Degree of openness × Contour type | 0.68 | 0.02 | 1 | 28 | 37.38 | <0.001 | 0.01 | 0.572 | 0.358~0.706 |
| Ceiling height × Degree of openness × Contour type | 0.75 | 0.02 | 1 | 28 | 31.78 | <0.001 | 0.013 | 0.532 | 0.309~0.677 |

**Table 2. ANOVA results for pleasantness ratings.**

| Factors | MS | MSE | df1 | df2 | F-statistic | Uncorrected p-value | Corrected p-value | $\eta^2 p$ | $\eta^2 p$ 90% CI |
|---|---|---|---|---|---|---|---|---|---|
| Ceiling height | 2.61 | 0.05 | 1 | 28 | 53.38 | <0.001 | 0.007 | 0.656 | 0.468~0.765 |
| Degree of openness | 5.04 | 0.05 | 1 | 28 | 109 | <0.001 | 0.008 | 0.796 | 0.672~0.862 |
| Contour type | 0.05 | 0.02 | 1 | 28 | 2.32 | 0.139 | 0.017 | 0.077 | 0.000~0.266 |
| Ceiling height × Degree of openness | 0 | 0.02 | 1 | 28 | 0.01 | 0.921 | 0.05 | 0 | 0.000~0.032 |
| Ceiling height × Contour type | 0 | 0.02 | 1 | 28 | 0.03 | 0.873 | 0.025 | 0.001 | 0.000~0.063 |
| Degree of openness × Contour type | 0.62 | 0.03 | 1 | 28 | 22.74 | <0.001 | 0.01 | 0.448 | 0.217~0.615 |
| Ceiling height × Degree of openness × Contour type | 1.28 | 0.02 | 1 | 28 | 63.48 | <0.001 | 0.013 | 0.694 | 0.521~0.792 |

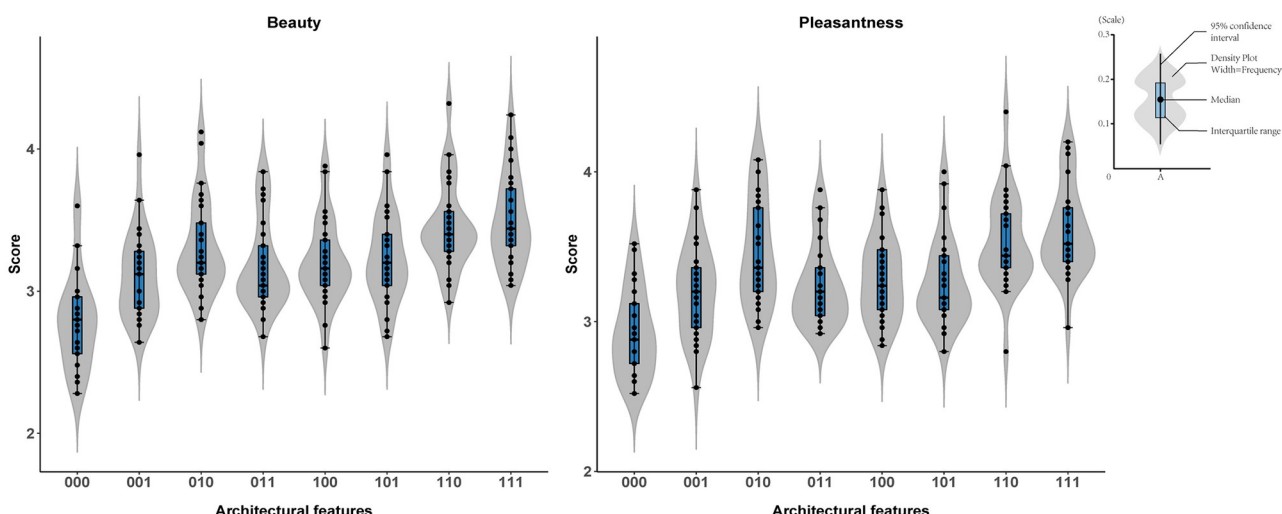

**Fig 3. Beauty and pleasantness rating.** The first digit in the x-axis represents the ceiling height, the second digit represents the degree of openness, and the third digit represents the contour type. In the violin plot, the box in the middle indicates the interquartile range, and the vertical line covers the 95% confidence interval.

the beauty and pleasantness rating. For architectural space with lower ceilings and less open space, curvilinear contours were rated higher than rectilinear contours (p<0.05). In the combination of lower ceilings and more open space, rectilinear contours were more likely to lead to a high beauty and pleasantness score (p<0.05). When buildings have lower ceilings and

**Table 3. Mean and SD of beauty and pleasantness ratings.**

| Hight (low = 0) | Openness (closed = 0) | Contour (rectilinear = 0) | Pleasantness | | Beauty | | N |
|---|---|---|---|---|---|---|---|
| | | | Mean | S.D. | Mean | S.D. | |
| 0 | 0 | 0 | 72.86 | 7.08 | 69.66 | 8.45 | 29 |
| 0 | 0 | 1 | 79.97 | 7.72 | 77.55 | 7.4 | 29 |
| 0 | 1 | 0 | 86.59 | 8.84 | 82.55 | 8.1 | 29 |
| 0 | 1 | 1 | 81.07 | 6.76 | 79.38 | 7.83 | 29 |
| 1 | 0 | 0 | 82 | 6.76 | 79.9 | 7.22 | 29 |
| 1 | 0 | 1 | 81.52 | 7.22 | 80.69 | 7.72 | 29 |
| 1 | 1 | 0 | 88.21 | 7.67 | 86.28 | 7.71 | 29 |
| 1 | 1 | 1 | 89.97 | 7.25 | 87.34 | 7.83 | 29 |

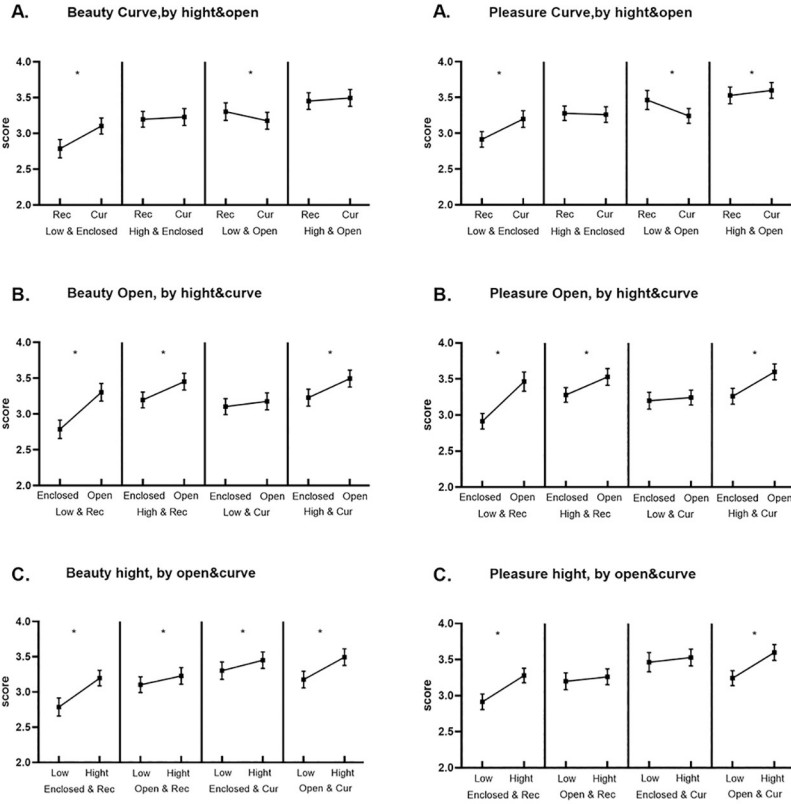

**Fig 4. Comparison between beauty and pleasantness ratings in different conditions.**

rectilinear contours or higher ceilings and rectilinear contours or higher ceilings and curvilinear contours, more open space is more likely to lead to a higher beauty and pleasantness score than is less open space ($p<0.05$). Among the 4 combinations, compared to lower ceilings, higher ceilings were more likely to lead to a higher beauty and pleasantness score ($p<0.05$).

We used PASS vesion 15.0 software to calculate the effectiveness power. based on our study data, after entering the values of the parameters required by the software such as sample size = 29, test level $\alpha$ = 0.05, $\rho$ (Autocorrelation) for each independent variable and the mean and standard deviation for each group, the power of each term of the model was calculated as Table 4.

**Table 4. Power analysis.**

| factors | Power(beauty) | Power(pleasantnes) |
|---|---|---|
| Hight | >0.999 | 0.997 |
| Openness | >0.999 | >0.999 |
| Contour | 0.496 | 0.149 |
| Hight×Openness | 0.112 | 0.051 |
| Hight×Contour | 0.368 | 0.054 |
| Openness×Contour | >0.999 | >0.999 |
| Hight×Openness×Contour | >0.999 | >0.999 |

## Discussion

The current results suggest that Chinese observers prefer architectural space with high ceilings and open space. The preference to curvilinear contours interacts with ceiling heights and openness. The preference to high ceilings, open space, and curvilinear contours has also been shown for Western observers [1, 10, 22]. Since the current study only employs Chinese observers as the participant, it cannot quantify whether the preference to architectural features varies across cultures. The current study find that the preference to curvilinear contours depends on the ceiling height and openness of the space. Future studies are needed to test whether Western observers also prefer curvilinear contours only when the ceiling is low and the space is enclosed. Although previous studies have not analyzed how the preference to contour relies on ceiling height and space openness, a recent study has shown that experience can strongly modulate preference to curvilinear contours [22]. The study shows that, within the Western culture, self-identified architects and designers show stronger preference to curvilinear contours than non-experts. In sum, combing the current results and previous results [1, 10, 22], it is shown that human observers prefer high ceilings and open space, and also prefer curvilinear contours in some conditions.

## Supporting information

**S1 Fig. Correlation between beauty and pleasantness ratings.**
(TIF)

**S1 File.**
(RAR)

**S1 Data.**
(RAR)

## Author Contributions

**Conceptualization:** Nai Ding.

**Data curation:** Anbang Dai, Jiajie Zou.

**Formal analysis:** Jiajie Zou.

**Investigation:** Anbang Dai, Junru Wang.

**Supervision:** Hiroatsu Fukuda.

**Validation:** Hiroatsu Fukuda.

**Writing – original draft:** Anbang Dai.

**Writing – review & editing:** Nai Ding, Hiroatsu Fukuda.

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
