## [Decision Letter · Decision Letter 0]

17 Aug 2021

PONE-D-21-20164

Cultural influence on the aesthetic judgment of architecture

PLOS ONE

Dear Dr. Dai,

Thank you for submitting your manuscript to PLOS ONE. After careful consideration, we feel that it has merit but does not fully meet PLOS ONE’s publication criteria as it currently stands. Therefore, we invite you to submit a revised version of the manuscript that addresses the points raised during the review process.

I encourage you to take both Reviewers' comments and suggestions very seriously, as they both are highly expert in this field and have raised quite a long list of valid criticisms that will need to be addressed in a revision. The theoretical issues raised by both reviewers will need particularly careful and nuanced handling (including Reviewer 2's issues with drawing strong conclusions about culture when no cultural comparison exists), and Reviewer 1 has provided a wealth of valuable critiques and insights about the theoretical framing as well as statistical reporting (and has highlighted some valuable additional literature that I strongly encourage you to include). The two expert reviewers and I will be carefully evaluating how each of their points have been addressed in this revision.

We look forward to receiving your revised manuscript.

Kind regards,

Emily S. Cross

Academic Editor

PLOS ONE

Reviewers' comments:

Reviewer's Responses to Questions

**Comments to the Author**

1. Is the manuscript technically sound, and do the data support the conclusions?

Reviewer #1: Partly

Reviewer #2: Partly

2. Has the statistical analysis been performed appropriately and rigorously? 

Reviewer #1: No

Reviewer #2: Yes

3. Have the authors made all data underlying the findings in their manuscript fully available?

Reviewer #1: No

Reviewer #2: Yes

4. Is the manuscript presented in an intelligible fashion and written in standard English?

Reviewer #1: Yes

Reviewer #2: Yes

5. Review Comments to the Author

Reviewer #1: The paper investigated whether Chinese participants prefer architectural spaces characterised by features, such as curvilinear contours, high ceilings and open spaces. The results showed that Chinese participants prefer buildings with high ceilings, open spaces and curvilinear contours. While the paper seeks to answer an interesting question, I believe it is lacking in theoretical and methodological details which prevents the findings from being clear and convincing. I outline these limitations below.

Introduction:

1. The introduction is lacking important detail and makes some unsupported claims.

For example, on page 1, lines 4-5, the authors discuss the impact of environmental characteristics on ‘neurological and physiological response in humans’, however, there are no references cited for the claims made.

Also, on page 1, lines 16-17, the reference cited (4) is not appropriate here. 'Vers une architecture' by Le Corbusier is a manifesto (a collection of essays) for a new architectural style based on function and not empirical studies.

Page 2 – first paragraph - the authors should provide more detail when summarising previous studies about the link between curvature, high ceilings and open spaces and aesthetic preference.

In this sense, the authors might consider including newer papers as well, such as

Coburn, A., Vartanian, O., Kenett, Y. N., Nadal, M., Hartung, F., Hayn-Leichsenring, G., … Chatterjee, A. (2020). Psychological and neural responses to architectural interiors. Cortex, 126, 217–241. https://doi.org/10.1016/j.cortex.2020.01.009

pre-print - Skov, M., Vartanian, O., Navarrete, G., Modroño, C., Chatterjee, A., Leder, H., … Nadal, M. (2021). Grey Matter Volume and architecture. Differences in Regional Grey Matter Volume Predict the Extent to which Openness influences Judgments of Beauty and Pleasantness of Interior Architectural Spaces. https://doi.org/10.1101/2021.03.31.437827

2. The authors should provide a clearer summary of the hypotheses and predictions at the end of the Introduction.

Method:

1. Please use research “participants” when referring to individuals who take part in psychological research, as it can be argued, it is more respectful of research volunteers.

2. Please state more clearly how many participants completed this study.

3. Were any participants excluded from the analyses? If yes, please mention the exclusion criteria.

4. The authors should offer a justification for the sample size used in this study.

5. For more clarity, it would be useful if the authors could provide a subsection "Stimuli and Procedure"

6. It would be useful if the authors could provide a graphical illustration of an experimental trial for a better understanding of the research procedure.

7. The authors should provide clearer details about the experimental procedure and the duration of the experiment. I find it confusing that the experiment length varied between 35 and 75 minutes. What is the justification for that?

Results:

1. It would be helpful to report the descriptive statistics - mean and SD for each condition.

2. It would be useful to mention whether the assumptions for ANOVA test have been met.

3. The results figures are lacking important detail. It would be helpful if individual data were plotted out. Violin plots would be more useful to see the spread and direction of the data. Also, they may help in identifying whether the data were normally distributed.

Please provide error bars 95% CI for all the figures.

4. It is very confusing why the authors have used the R results output tables instead of reporting all the F statements for main effects and interactions. It would be useful to report all the F statements in the results section.

5. Also, the results for the post hoc comparisons should be reported in the results section.

6. A table containing the mean differences across conditions would be helpful.

7. Please provide your data and analysis pipeline online if possible.

Discussion:

1. Please interpret your results in accordance with the main key hypotheses.

2. More caution needs to be exerted when interpreting the results of this study. I find it confusing and misleading that the authors interpret the current findings as a cultural variation assessment in aesthetic preference of architectural features. This is not quite right, as the current study did not use participants from both Eastern and Western cultures so that to really compare the two cultures and to provide support for the cultural aesthetic variation claim.

However, the current findings can be linked to previous studies conducted in the Western culture, but the methodological differences between the current study and previous studies should be clearly stated (e.g., sample size, experimental procedure, statistical analyses used). This aspect should be considered in both discussion and abstract.

3. The authors should address the potential limitations of this study in a thorough manner, especially regarding the small sample size and statistical power.

Reviewer #2: The manuscript presents an interesting study comparing the effects of contours, high ceilings and open spaces in interior environments on ratings of beauty and pleasantness. The study aims to replicate parts of a previous study (Vartanian, O., Navarrete, G., Chatterjee, A., Fich, L. B., Leder, H., Modroño, C., ... & Skov, M. (2013). Impact of contour on aesthetic judgments and approach-avoidance decisions in architecture. Proceedings of the National Academy of Sciences, 110(Supplement 2), 10446-10453.) but using Chinese participants rather than Western (Spanish) participants.

The title of the manuscript suggests the study examines cultural influence on the aesthetic judgement of architectural design. However, I think only very tentative conclusions can be drawn regarding culture. To make inferences about cultural influence, a direct comparison between participants from different cultures should have been carried out. This was not the case, only participants from an Eastern culture were used. Comparison against the results from an entirely different study (admittedly using the same stimuli, but no further details are given about the similarities between the two studies) can not provide conclusive evidence about cultural influences. Any differences between the two studies could have been due to a range of different reasons, other than culture.

Another major issue I have with the manuscript is the lack of reflection on previous findings and how these relate to the findings reported in the study carried out by the authors. A large part of the Discussion section (e.g. lines 193-208) talks about potential differences between Western and Eastern perceptions of architecture, but there is virtually no discussion prior to this comparing the existing results with those found in previous literature, particularly in the study that this work is based on (Vartanian et al). I would like to see a more detailed description and comparative analysis carried out with the previous related literature. The authors state that “After comparing the results of this study with those of previous studies, we postulate that differences in viewers’ perceptions...is potentially related to cultural background” (lines 190192). However there is little to no evidence of this comparison of results.

Other issues for the authors to consider:

A key limitation of the study that should be acknowledged is that the factors of interest couldn't be manipulated whilst keeping other variables constant. This should be acknowledged in the Discussion. It is possible that the factors investigated varied with some other unmeasured confounding variable/s.

Line 17 - reference [4] does not seem appropriate here.

Some justification of sample size would be useful. Was no power analysis carried out based on the findings of the previous study (reference [5])?

The pictures may have been from reference [5] (line 58), but were they classified in the same way (i.e. into the sub-groups if high/low ceiling, open/closed, and curvilinear/rectilinear? Please clarify. It is important to know whether you were using the same classifications as in the earlier study, or whether you carried out your own classification process.

Line 80 - reference missing.

It’s not clear why the total score for each set of images is used (giving a range of 25-125 for each image), rather than just take the mean score, which would be more intuitive as it would fit with the response scale used.

The authors state that simple effect comparisons do not need p-value adjustment (line 105), but p-values in the ANOVA do require adjustment.

It is helpful that the raw data has been provided in the supplementary file, however this data needs some kind of codebook or readme file to explain how to understand and use it. Ideally the data for each participant would be put on one spreadsheet / csv file for ease of use, and definitions / explanations given for each variable. Furthermore, it would be useful to provide the analysis script as well, as a supplementary file.

Lines 135-136 refer to pleasantness score, but should this be beauty score? The analysis of pleasantness appears to begin from line 138.

The captions for Fig 2 and 4 include reference to 3 different digits, this is confusing and does not seem to match with the actual figures.

The footnote to tables 1 and 2 refer to effect size thresholds, and also that 90% CI is reported rather than 95% CI. Do you have references to support these statements?

S1 Figure shows the correlations between pleasantness and beauty. These are very strong - did you consider just combining the two ratings into one overall ‘aesthetic’ score? To analyse both separately seems unnecessary.

6. PLOS authors have the option to publish the peer review history of their article (what does this mean?). If published, this will include your full peer review and any attached files.

Reviewer #1: No

Reviewer #2: **Yes: **Jim Uttley

---

## [Decision Letter · Decision Letter 1]

29 Nov 2021

PONE-D-21-20164R1Aesthetic Judgment of Architecture for Chinese ObserversPLOS ONE

Dear Dr. Dai,

Thank you for submitting your manuscript to PLOS ONE. After careful consideration, we feel that it has merit but does not fully meet PLOS ONE’s publication criteria as it currently stands. Therefore, we invite you to submit a revised version of the manuscript that addresses the points raised during the review process.

Both reviewers have raised detailed and very, very helpful suggestions for further improving this manuscript before it is acceptable for publication. As only minor revisions are being requested at this stage, I would like to strongly encourage the authors to carefully revise the manuscript in line with both reviewers' suggestions before resubmitting.

We look forward to receiving your revised manuscript.

Kind regards,

Emily S. Cross

Academic Editor

PLOS ONE

Journal Requirements:

Reviewers' comments:

Reviewer's Responses to Questions

**Comments to the Author**

1. If the authors have adequately addressed your comments raised in a previous round of review and you feel that this manuscript is now acceptable for publication, you may indicate that here to bypass the “Comments to the Author” section, enter your conflict of interest statement in the “Confidential to Editor” section, and submit your "Accept" recommendation.

Reviewer #1: (No Response)

Reviewer #2: (No Response)

2. Is the manuscript technically sound, and do the data support the conclusions?

Reviewer #1: Partly

Reviewer #2: Partly

3. Has the statistical analysis been performed appropriately and rigorously? 

Reviewer #1: Yes

Reviewer #2: No

4. Have the authors made all data underlying the findings in their manuscript fully available?

Reviewer #1: Yes

Reviewer #2: No

5. Is the manuscript presented in an intelligible fashion and written in standard English?

Reviewer #1: Yes

Reviewer #2: Yes

6. Review Comments to the Author

Reviewer #1: The manuscript has been considerably improved, however, there are still some pending issues, which I outline below.

Introduction

- Citation needed at the end of this sentence - ‘Once a certain architectural element fits in a certain life scene, such as work, study, and rest, it can enhance behavioral effects through positive emotions.’

- I would suggest providing more evaluation and interpretation of the previous studies. For example, ‘it is suggested that the influence of openness on pleasantness and beauty involves in the anterior prefrontal cortex and the temporal pole.’ This is interesting, however, without mentioning some functional aspects, such as the role of the anterior temporal lobe in semantic memory and how that might relate to drawing meaning from our environment, the text is not convincing enough.

- In the last paragraph of the introduction, I would suggest clearly stating that this study was conducted in Chinese participants only. I find it confusing that the authors still use a comparison term with Western participants although they did not test Western participants.

Methods

- Although the authors claim they did refer now to participants rather than to subjects, the term ‘subjects’ still appears 15 times in the method section.

- ‘after completing the experiment each participant received a reward’ – Please provide more details about the nature of the reward received by participants.

- ‘Candidate participants who had knowledge about the design and process of this experiment were excluded’ – this is confusing, please mention how many participants were excluded from analyses.

- although the figure explaining the experimental procedure is very helpful, it is not clear whether the pictures were randomly presented or whether the order of different sets of pictures was fixed across participants.

- Citation needed when referring to Matlab and Psychtoolbox - ‘The experimental program was written using the MATLAB 2018 psychtoolbox software’.

Results

- although the authors have included a violin plot, it lacks important detail. It would be helpful if individual data were plotted out. At the same time, it is unclear which line represents 95% CI. Also, the text describing the conditions for the x-axis is very busy. I would suggest using a shortened name for conditions, background grid lines, individual data points plotted out and adding a legend explaining thoroughly the figure.

- In the current version of the manuscript, the effect size (partial eta squared) is missing. Please report the effect size.

- Are there any reasons for not including the results for pleasantness ratings in the main results section? - ‘The results for pleasantness rating were quantitatively similar to the results for beauty rating (Fig. S1) and were shown in Table. S1, Figs. S2 and S3’. I find it confusing for deciding to place those on the supplementary material, especially as in the discussion, the authors refer to both beauty and pleasantness rating results. Moreover, Vartanian et al., 2013 reported both pleasantness and beauty ratings in the main results section.

Discussion

- The first and the last paragraphs in the discussion should make it unequivocally clear that this study was conducted with Chinese participants. As it stands, it is implying a Western comparison experimental condition, which is not true.

- Paragraph two – citation needed after - ‘Preference for high ceilings and open space are consistent with previous results.’

- I do not think it is useful to report p-values in the discussion section. I would suggest discussing how the results either support or do not support the hypotheses.

Reviewer #2: I am grateful to the authors for their responses to my comments. The majority of my comments have been addressed adequately. However some responses do not fully address my original concerns.

Response to comment 3) - the additional discussion of limitation does not sufficiently mention the fact that the study was unable to change a factor of interest whilst keeping other factors constant. This introduces potential confounds and should therefore be mentioned clearly as a limitation. I do not think the authors’ current text does this.

Response to comment 4) - just mentioning what the previous study’s sample size was is not the same as a power analysis. It would be useful to understand what effect sizes could have been detected with your sample size, and how these compared to the effect sizes that might be expected based on the previous Vartanian et al study.

Response to comment 5) - the text in the data analysis section still refers to the sum of the scores being calculated, with a value range of 25-125. This could be confusing, if you are now presenting mean values in the figures etc.

Response to comment 6) - the p-values in an ANOVA test still require adjustment to account for inflated Type I error. For example see this paper: Cramer, A. O., van Ravenzwaaij, D., Matzke, D., Steingroever, H., Wetzels, R., Grasman, R. P., ... & Wagenmakers, E. J. (2016). Hidden multiplicity in exploratory multiway ANOVA: Prevalence and remedies. Psychonomic bulletin & review, 23(2), 640-647.

Response to comment 7) - it would be more helpful to include a better-described set of data, and analysis code, with the submission rather than after acceptance. This would ensure the data and its description / use is intrinsically part of the paper.

7. PLOS authors have the option to publish the peer review history of their article (what does this mean?). If published, this will include your full peer review and any attached files.

Reviewer #1: No

Reviewer #2: **Yes: **Jim Uttley

---

## [Author Response · Author response to Decision Letter 1]

29 Dec 2021

We would like to thank the reviewers for their constructive suggestions. Please see below for our point-to-point replies.

REVISIONS FOR THIS PAPER:

Introduction

- Citation needed at the end of this sentence - ‘Once a certain architectural element fits in a certain life scene, such as work, study, and rest, it can enhance behavioral effects through positive emotions.’

We have added references for the statements.

“Once a certain architectural element fits in a certain life scene, such as work, study, and rest, it can enhance behavioral effects through positive emotions [7, 8] ”

- I would suggest providing more evaluation and interpretation of the previous studies. For example, ‘it is suggested that the influence of openness on pleasantness and beauty involves in the anterior prefrontal cortex and the temporal pole.’ This is interesting, however, without mentioning some functional aspects, such as the role of the anterior temporal lobe in semantic memory and how that might relate to drawing meaning from our environment, the text is not convincing enough.

We have removed the discussion about neuroimaging results since the current study only analyzed behavior.

- In the last paragraph of the introduction, I would suggest clearly stating that this study was conducted in Chinese participants only. I find it confusing that the authors still use a comparison term with Western participants although they did not test Western participants.

We have clearly mentioned that the study only tested Chinese participants:

“If the preference to architectural features is strongly influenced by daily architectural aesthetic experience, observers living in environments with different building styles may prefer different architectural features [20-22]. Here, we analyze the preference to architectural features, including ceiling height, openness, and contour, in Chinese observers.”

We have now also clearly mentioned in the discussion that we do not compare our study with the previous studies.

“The current results suggest that Chinese observers prefer architectural space with high ceilings and open space. The preference to curvilinear contours interacts with ceiling heights and openness. The preference to high ceilings, open space, and curvilinear contours has also been shown for Western observers [1, 10, 22]. Since the current study only employs Chinese observers as the participant, it cannot quantify whether the preference to architectural features varies across cultures.”

Methods

- Although the authors claim they did refer now to participants rather than to subjects, the term ‘subjects’ still appears 15 times in the method section.

We have made sure that “subjects” are now replaced with “participants”.

- ‘after completing the experiment each participant received a reward’ – Please provide more details about the nature of the reward received by participants.

We have added more details:

“Each participant received 40 RMB monetary reward”

- ‘Candidate participants who had knowledge about the design and process of this experiment were excluded’ – this is confusing, please mention how many participants were excluded from analyses.

The first and second authors were the first participants to test the experiment program. Since the authors knew previous studies and might have potential biases, their data were not analyzed. We agree that the statement is confusing and it is not necessary to report program testing. Therefore, the sentence is now removed. 

- although the figure explaining the experimental procedure is very helpful, it is not clear whether the pictures were randomly presented or whether the order of different sets of pictures was fixed across participants.

The 200 pictures are presented in a randomized order for each participant. We have now mentioned:

“The order of pictures was randomized for each participant.”

- Citation needed when referring to Matlab and Psychtoolbox - ‘The experimental program was written using the MATLAB 2018 psychtoolbox software’.

A reference has been added.

Results

- although the authors have included a violin plot, it lacks important detail. It would be helpful if individual data were plotted out. At the same time, it is unclear which line represents 95% CI. Also, the text describing the conditions for the x-axis is very busy. I would suggest using a shortened name for conditions, background grid lines, individual data points plotted out and adding a legend explaining thoroughly the figure.

We have added individual data, and added more details to the figure and figure caption.

- In the current version of the manuscript, the effect size (partial eta squared) is missing. Please report the effect size.

Effect size has been added to Table 1 and Table 2.

- Are there any reasons for not including the results for pleasantness ratings in the main results section? - ‘The results for pleasantness rating were quantitatively similar to the results for beauty rating (Fig. S1) and were shown in Table. S1, Figs. S2 and S3’. I find it confusing for deciding to place those on the supplementary material, especially as in the discussion, the authors refer to both beauty and pleasantness rating results. Moreover, Vartanian et al., 2013 reported both pleasantness and beauty ratings in the main results section.

We have moved the pleasantness results back to the main results section.

Discussion

- The first and the last paragraphs in the discussion should make it unequivocally clear that this study was conducted with Chinese participants. As it stands, it is implying a Western comparison experimental condition, which is not true.

We have modified the result section:

“The current results suggest that Chinese observers prefer architectural space with high ceilings and open space. The preference to curvilinear contours interacts with ceiling heights and openness. The preference to high ceilings, open space, and curvilinear contours has also been shown for Western observers [1, 10, 22]. Since the current study only employs Chinese observers as the participant, it cannot quantify whether the preference to architectural features varies across cultures.”

- Paragraph two – citation needed after - ‘Preference for high ceilings and open space are consistent with previous results.’

The previous study refers to the study by Vartanian et al. Since a similar statement has already been made earlier, this sentence is now removed.

- I do not think it is useful to report p-values in the discussion section. I would suggest discussing how the results either support or do not support the hypotheses.

We have removed p-values in the discussion section.

Reviewer #2: I am grateful to the authors for their responses to my comments. The majority of my comments have been addressed adequately. However some responses do not fully address my original concerns.

Response to comment 3) - the additional discussion of limitation does not sufficiently mention the fact that the study was unable to change a factor of interest whilst keeping other factors constant. This introduces potential confounds and should therefore be mentioned clearly as a limitation. I do not think the authors’ current text does this.

We have thoroughly modified the manuscript to make sure that we no longer mention that we compare the results across studies. For example, in the discussion section, it is mentioned that 

“The current results suggest that Chinese observers prefer architectural space with high ceilings and open space. The preference to curvilinear contours interacts with ceiling heights and openness. The preference to high ceilings, open space, and curvilinear contours has also been shown for Western observers [1, 10, 22]. Since the current study only employs Chinese observers as the participant, it cannot quantify whether the preference to architectural features varies across cultures.”

Response to comment 4) - just mentioning what the previous study’s sample size was is not the same as a power analysis. It would be useful to understand what effect sizes could have been detected with your sample size, and how these compared to the effect sizes that might be expected based on the previous Vartanian et al study.

We have now reported the effect size of the current study in Tables 1 and 2. We did not calculate the effect size for the Vartanian et al., study since the data are not publically available. None of the previous studies using the same picture set have reported the effect size, although the trend is consistent across studies [1] [10] [22] [29]. More importantly, we no longer compare our results with the Vartanian results.

Response to comment 5) - the text in the data analysis section still refers to the sum of the scores being calculated, with a value range of 25-125. This could be confusing, if you are now presenting mean values in the figures etc.

We have now consistently reported the mean values in figures and tables. 

Response to comment 6) - the p-values in an ANOVA test still require adjustment to account for inflated Type I error. For example see this paper: Cramer, A. O., van Ravenzwaaij, D., Matzke, D., Steingroever, H., Wetzels, R., Grasman, R. P., ... & Wagenmakers, E. J. (2016). Hidden multiplicity in exploratory multiway ANOVA: Prevalence and remedies. Psychonomic bulletin & review, 23(2), 640-647.

We have now reported corrected P-value in Tables 1 and 2.

“The effect of multiple comparisons [25] were corrected using Bonferroni correction.”

Response to comment 7) - it would be more helpful to include a better-described set of data, and analysis code, with the submission rather than after acceptance. This would ensure the data and its description / use is intrinsically part of the paper.

We have uploaded the data and code.

---

## [Editor Report · Decision Letter 2]

2 Mar 2022

Aesthetic Judgment of Architecture for Chinese Observers

PONE-D-21-20164R2

Dear Dr. Dai,

We’re pleased to inform you that your manuscript has been judged scientifically suitable for publication and will be formally accepted for publication once it meets all outstanding technical requirements.

Kind regards,

Emily S. Cross

Academic Editor

PLOS ONE

Additional Editor Comments (optional):

While I very much appreciate the attention to detail that you and your coauthors paid to the reviewers' detailed comments, I did find the updated discussion on the brief side - but I think sticking to your data and allowing readers to draw more of their own conclusions based on your (now much more clearly reported) findings is no bad thing. Congratulations once more on your paper acceptance.

---

## [Editor Report · Acceptance letter]

21 Mar 2022

PONE-D-21-20164R2 

Aesthetic Judgment of Architecture for Chinese Observers 

Dear Dr. Dai:

I'm pleased to inform you that your manuscript has been deemed suitable for publication in PLOS ONE. Congratulations! Your manuscript is now with our production department. 

Kind regards, 

on behalf of

Professor Emily S. Cross 

Academic Editor

PLOS ONE